

# Knowledge and attitudes towards stem cells and the significance of their medical application among healthcare sciences students of Jouf University

Abdulrahman Almaeen[1], Farooq Ahmed Wani[2] and Ashokkumar Thirunavukkarasu[3]

[1] Department of Pathology, College of Medicine, Jouf University, Sakaka, Aljouf, Saudi Arabia
[2] Department of Pathology, College of Medicine, Jouf University, Sakaka, Aljouf, Saudi Arabia
[3] Department of Community Medicine, College of Medicine, Jouf University, Sakaka, Aljouf, Saudi Arabia

Corresponding author
Farooq Ahmed Wani,
fawani@ju.edu.sa

## ABSTRACT

**Background**. Adequate knowledge and positive attitude of the medical and dental students towards the stem cells and their utilization in medical science is extremely important keeping in view the ever-increasing potential of stem cells in the medical field. The present study was planned to assess the knowledge and attitude of the medical and dental students towards stem cells and their applications in medical science.

**Methods**. This cross-sectional study was conducted among 217 medical and dental college students of the Jouf University. The systematic random sampling method was used to select students based on gender and year of study. After obtaining written informed consent, a self-administered questionnaire consisting of three parts was administered to the students. The first part collected the socio-demographic details; part 2 contains 15 questions regarding knowledge and part 3 contains 10 questions regarding attitude towards stem cells.

**Results**. Majority of the participants were males (54.4%) in the age group of 21–22 years. Awareness regarding Saudi stem cell donor registry was observed in 50.7% of the students . A total of 72.4% of the students possessed medium knowledge while 70% of the students possessed high attitude score towards stem cell research and its medical significance. A significant relationship was observed between the Saudi Stem Cell Donor Registry awareness and knowledge score with p-value of 0.04. Relationship between the knowledge and attitude scores was significant with p-value of 0.001 and with a Pearson correlation score of $r = 0.334$.

**Conclusion**. Medium to high level of knowledge was noted among majority of the participants and a high attitude score was also noted towards stem cells and their relevance. A significant positive correlation was observed between the knowledge and attitude scores. It is recommended to include various interventional educational programs for the medical and dental students on the significance of stem cells in the medical field.

## INTRODUCTION

Stem cells are characterized by the intrinsic ability of perpetual self-renewal and can differentiate into specific cell types. Stem cells in general are classified into embryonic and adult stem cells. Embryonic stem cells are pluripotent signifying thereby their ability to differentiate into all cell types of the body. On the other hand, adult stem cells possess only restricted potential of differentiation and can differentiate into the cell types of their tissue of origin (*National Institutes of Health, 2019*).

Stem cells not only play a crucial role in tissue and organ maintenance by replenishing the dying cells but also help to repair and regenerate the damaged tissues (*Fujimaki et al., 2013*). Stem cell therapy is of utmost importance for treating and curing many hematological malignances. Stem cell therapy promises hope for many incurable diseases like neurodegenerative diseases, diabetes, cardiac disease, etc. but it is still in the experimental phase (*Ray et al., 2015*). Promising results have been found in the clinical trials when stem cells are used for treating multiple sclerosis (*Muraro et al., 2017*), type I diabetes, and spinal cord injury (*Prentice, 2012*).

Embryonic stem cells can be directed to differentiate along many different cell lines and appear to be very promising to treat different types of diseases but such research is associated with legal, ethical and social implications (*Pandey, Kaur & Kamath, 2016*). Umbilical cord blood (UCB) stem cells obtained from the blood in the umbilical cord and placenta are being preferred because of high proliferative potential, low antigenicity, decreased incidence of graft-versus-host reaction, without ethical implications and can be transplanted even without an identical HLA match (*Habib & Gordon, 2006*; *Ikuta, 2008*).

Adult stem cells may be used for autologous or allogeneic transplantation for the treatment of hematologic malignancies and degenerative disorders. The risk of graft rejection is avoided in autologous transplantation (*Dimarakis, Habib & Gordon, 2005*). Promising results have been found in clinical trials involving the regeneration of cardiac muscle (*Gordon et al., 2006*) and liver tissue (*Levičar et al., 2008*).

Medical and allied medical students are expected to have reasonable knowledge regarding stem cells and possess a positive attitude regarding their application in medicine. Variable results have been found in the different studies conducted throughout the world. *Lye et al. (2015)* showed that the nursing students possessed moderate knowledge and a positive attitude towards stem cell therapy. One of the studies conducted in Tabuk, Saudi Arabia revealed sub-optimal knowledge, attitude, and practice among the final class medical students and doctors regarding the use of stem cells in the management of diabetes (*ALtemani et al., 2018*). An educational interventional study on stem therapy done in Saudi Arabia on nursing students found poor knowledge but a positive attitude prior to the intervention. However after educational intervention, a significant increase in the level of knowledge was observed (*Azzazy & Mohamed, 2016*; *Walker et al., 2012*). Throughout the World, public cord blood banks (CBBs) store more than 700,000 UCB units. There are 2 public CBBs in Saudi Arabia but no private CCBs are present (*Jawdat et al., 2018*). Most studies have shown inadequate knowledge and awareness among public and healthcare staff about CB banking (*Perlow, 2006*; *Suen & Lao, 2011*; *Tuteja, Agarwal & Phadke, 2016*).

The present study was planned to assess the knowledge and attitude of the students of Medical College and Dental College towards stem cells and their applications in medical science and to correlate knowledge and attitude scores among them.

## METHODOLOGY

### Study design and settings

This cross-sectional study was conducted among the medical and dental college students of Jouf University. Jouf University is situated in the Sakaka region of Aljouf province in the Kingdom of Saudi Arabia. Currently, this is the only university serving in this region. The total registered student in both medical and dental colleges of the Jouf University in the year 2019–20 is 662.

### Sample size estimation

The sample size was calculated based on the Cochrane's formula of sample size estimation ($n = z^2 pq/e^2$), where n is the minimum required sample size, z is desired confidence interval (95%), p is the proportion, $q = 1 - p$ and e is the desired level of precision (5%). We have taken 50% as population proportion (p) to estimate the sample size for the present study. Applying the above formula and values with the finite population of 662, the estimated sample size was calculated as 244.

### Sampling method

A multistage probability proportional to size (PPS) sampling method was applied to select the study participants. Firstly, the required number of students from each college was selected based on the total number of registered students from that college. In the next stages, the required fractions of participants from each year, and gender also were selected based on PPS. By using the university identification number of each student, the systematic random sampling method was used to select from each gender and year.

### Ethical consideration

The Local Committee for Bioethics (LCBE) of Jouf University had given ethical clearance to conduct this study (Approval no: 09-02/41). The data was collected from the participants after their willingness to participate given through written informed consent.

### Data collection procedure

After completion of necessary approval formalities, the data collection process was initiated. The selected students were communicated through their class leader on their availability for data collection. Data collection during examination time was avoided as students may not be available. All the registered students of the medical and dental college of the academic year 2019–20 were included in the study. The research team made three attempts in one month to communicate with the selected students. Those students who could not be contacted by the principal and co-investigators despite three attempts in one month or those participants who were not willing to participate were considered as non-respondent.

## Data collection instrument

After obtaining written informed consent, a paper form of self-administered questionnaire was given to study participants. This questionnaire was adapted from a previously published study by *Lye et al. (2015)*. The scale used in their study has the Cronbach alpha of 0.67 for knowledge section and 0.86 for attitude section. A pilot study was done among 30 medical and dental college students of Jouf University using the above adapted questionnaire. The results from pilot study showed the Cronbach alpha of 0.71 for knowledge section and 0.81 for attitude section, which exhibited good internal consistency in the format used. Hence, we proceeded with the adapted questionnaire and it consisted of three parts as mentioned below. The first part collected the socio-demographic details such as age, gender, college type, year of education, marital status, and smoking behavior of study population. This section also inquired about participants' awareness about Saudi Stem Cells Donor Registry. In part 2, participants responded to 15 items that assessed their knowledge of stem cells and their uses in medical sciences. Selected respondents had to choose yes or no for each item. The responses yes were given as 2 marks and no was given as 1 mark. Hence the expected range of scores is 15 to 30. The obtained knowledge scores were divided into three categories namely low (15–20), medium (21–25), and high (26–30). Part 3 consists of 10 items and responses were recorded on a 5-point Likert scale: Strongly agree, agree, not sure, disagree, and strongly disagree. Scores of responses were given; five for strongly agree to one for strongly disagree. Since there were 10 items in attitude sections, the range of scores varied from ten to fifty. The obtained attitude scores were divided into four categories namely low (10–20), medium (21–30), good (31–40), and excellent (41–50).

## Statistical analysis

Statistical package for social sciences (SPSS, USA) version 21.0 were used for data entry and analysis. Descriptive data of the study were presented as frequency (n) and percentage and quantitative data were presented as mean and standard deviation (SD). The data were checked for normal distribution through kurtosis -skewness analysis and Kolmogorov–Smirnov test. The kurtosis-skewness values of the sample met the normality assumption of data (Kurtosis and skewness value of knowledge score were 0.148 and 0.001, and for attitude score were −0.350 and −0.064). The statistical values of Kolmogorov–Smirnov tests of knowledge and attitude scores were 0.111 and 0.059 respectively. Independent $t$-test and one-way analysis of variance (ANOVA) were applied to compare the knowledge and attitude mean scores with the different socio-demographic characteristics of the study population as a hypothetical test. Chi-square test was done to find the association between different socio-demographic characteristics of the study population and their awareness about Saudi Stem Cell Donor Registry (SSCDR). Pearson's correlation test was applied to find the correlation between knowledge and attitude scores. A correlation coefficient value was interpreted as follows: less than 0.30 is a negligible correlation, 0.30 to 0.50 is a weakly positive correlation, 0.51 to 0.70 is a moderately positive correlation, 0.71 to 0.90 is a high positive correlation and more than 0.90 is a very high positive correlation (*Freedman, Pisani & Purves, 2007*). A $p$-value of less than 0.05 was considered a statistically significant

association between independent and outcome variables. All the statistical tests applied in the present study were two-tailed.

## RESULTS

The majority of the participants in our study were males (54.4%) and in the age group 21–22 years. A total of 61.3% of the students were from the Medical College and maximum (27.2%) were studying in 3rd year. Only 7.4% of the students were married. 12.4% of the participants were daily smokers (Table 1).

A total of 157 (72.4%) of the students possessed medium knowledge while 23% had high level of knowledge regarding stem cells. The majority (70%) of the students possessed high attitude score towards stem cell research and its medical significance whereas 18.9% had excellent attitude score (Table 2).

The relationship between the age, gender, college type, year of education, marital status and smoking status with the knowledge and attitude scores was not found to be significant (Table 3). The relationship between the SSCDR Awareness and knowledge score was significant with $p$-value of 0.041 whereas it was insignificant ($p$-value of 0.158) when it was compared with attitude score (Table 3).

A significant positive correlation ($r = 0.334$, $p < 0.001$) was obtained between the knowledge and attitude scores.

A significant association was observed between the SSCDR and gender of the participants ($p$-value of 0.020) and their smoking status ($p$-value of 0.002) (Table 4).

## DISCUSSION

Stem cells and their derived products offer considerable hope and assurance in devising new medical treatments. Stem cells play a great role in the field of transplant and regenerative medicine. In stem cell transplants, stem cells are used to replace the cells that have been destroyed by diseases and chemotherapy and have extremely beneficial effects in achieving remissions and cure in diseases like leukemias, lymphomas, and multiple myeloma. In regenerative medicine, they have immense potential of promoting the repair response in diseased, dysfunctional or injured tissue (*MayoClinic, 2019*). The knowledge and attitude of health care providers regarding stem cells is extremely important as they provide a reliable source of information to the patients which in turn improves the decision making power of patients regarding use of stem cells as a novel and innovative method of treatment (*Tork et al., 2017*). Health care professionals are considered as the most trustworthy, dependable and reliable source of information about stem cells and umbilical cord blood banking by the members of public (*Dinç & Şahin, 2009*; *Perlow, 2006*; *Rucinski et al., 2010*; *Venugopal et al., 2016*).

The majority of the participants in our study were males (54.4%) in the age group of 21–22 years. Nearly 2/3rd participants were Medical College students with 27.2% studying in the 3rd year. Only 7.4% of the students in our study were married whereas 12.4% were daily smokers (Table 1). *Lye et al. (2015)* in a study on undergraduate nursing students observed majority of the students were female (93.2%) from 3rd year (34%); maximum

**Table 1  Socio-demographic characteristics of the students (n = 217).**

| Variable | Frequency | Percentage |
|---|---|---|
| Age (mean ± SD) | 21.64 ± 1.65 | |
| Less than 21 years | 62 | 28.6 |
| 21–22 years | 86 | 39.6 |
| Above 22 years | 69 | 31.8 |
| Gender | | |
| Male | 118 | 54.4 |
| Female | 99 | 45.6 |
| College type | | |
| Medical | 133 | 61.3 |
| Dental | 84 | 38.7 |
| Year of education | | |
| 1st | 55 | 25.3 |
| 2nd | 48 | 22.1 |
| 3rd | 59 | 27.2 |
| 4th | 28 | 12.9 |
| 5th | 27 | 12.4 |
| Marital status | | |
| Single | 201 | 92.6 |
| Married | 16 | 7.4 |
| Smoking status | | |
| Daily | 27 | 12.4 |
| Rarely | 28 | 12.9 |
| Never | 162 | 74.7 |
| Awareness about Saudi Stem Cells Donor Registry | | |
| Yes | 110 | 50.7 |
| No | 107 | 49.3 |

**Table 2  Knowledge, and attitude of the students towards stem cell research and its medical significance (n = 217).**

| Category | Frequency | Percent |
|---|---|---|
| Knowledge | | |
| High | 50 | 23.0 |
| Medium | 157 | 72.4 |
| Low | 10 | 4.6 |
| Overall mean ± SD | 24.04 ± 2.13 | |
| Attitude | | |
| Excellent | 41 | 18.9 |
| High | 152 | 70.0 |
| Medium | 23 | 10.6 |
| Low | 0 | 0 |
| Overall mean ± SD | 36.63 ± 4.67 | |

**Table 3  Comparison of the mean (± SD) of study participants for socio-demographic characteristics.**

| Variables | Knowledge | | Attitude | |
|---|---|---|---|---|
| | Mean ± SD | p value | Mean ± SD | p value |
| Age category[*] | | | | |
| Less than 21 years | 23.71 ± 2.29 | 0.178 | 37.00 ± 4.99 | 0.680 |
| 21–22 years | 24.35 ± 2.05 | | 37.22 ± 4.66 | |
| Above 22 years | 23.94 ± 2.04 | | 35.57 ± 4.24 | |
| Gender[**] | | | | |
| Male | 23.80 ± 2.06 | 0.070 | 36.25 ± 4.41 | 0.195 |
| Female | 24.32 ± 2.18 | | 37.08 ± 4.94 | |
| College type[**] | | | | |
| Medical | 24.02 ± 1.95 | 0.850 | 36.64 ± 4.44 | 0.974 |
| Dental | 24.07 ± 2.39 | | 36.62 ± 5.04 | |
| Year of education[*] | | | | |
| 1st | 23.87 ± 2.29 | 0.344 | 36.35 ± 4.83 | 0.507 |
| 2nd | 24.25 ± 2.16 | | 37.96 ± 4.33 | |
| 3rd | 24.14 ± 2.05 | | 36.58 ± 4.90 | |
| 4th | 23.39 ± 2.20 | | 35.18 ± 4.83 | |
| 5th | 24.44 ± 1.72 | | 36.48 ± 3.91 | |
| Marital status[**] | | | | |
| Single | 24.01 ± 2.15 | 0.429 | 36.49 ± 4.74 | 0.121 |
| Married | 24.38 ± 1.78 | | 38.38 ± 3.36 | |
| Smoking status[**] | | | | |
| Daily | 23.44 ± 1.76 | 0.300 | 36.11 ± 3.86 | 0.200 |
| Rarely | 24.07 ± 2.09 | | 36.82 ± 4.64 | |
| Never | 24.13 ± 2.13 | | 36.69 ± 4.84 | |
| Stem cell Donor Registry Awareness[**] | | | | |
| Yes | 24.44 ± 2.06 | 0.041[***] | 37.07 ± 4.94 | 0.158 |
| No | 23.74 + 2.16 | | 36.18 ± 4.35 | |

**Notes.**
[*]One-way Analysis of Variance (ANOVA).
[**]Independent t-test.
[***]Statistically significant at the level of 0.05 (two-tailed).

with matriculation (81.8%) as their qualification and Islam (97.7%) was the predominant religion. *Jawdat et al. (2018)* in a study on public awareness of cord blood banking noted female predominance (88%), mostly college graduates (57%); and maximum participants (26%) were in the age group 19–25 years and belonged to the middle class socioeconomic status (82%).

The majority of the students possessed medium to high level of knowledge whereas poor knowledge was observed only in 4.6% of the students (Table 2). *Lye et al. (2015)* noted in their study that 92% of the nursing students had moderate knowledge score. *Venugopal et al. (2016)* observed in their study that most of the nurses' possessed good knowledge (42.86%) with a mean knowledge score of 16.84 ± 4.59 (*Venugopal et al., 2016*). We came across one study done by *Altemani et al. (2018)* on doctors and medical students in the Tabuk city regarding the knowledge of stem cell therapy in diabetes mellitus in

**Table 4 Association between socio-demographic characteristics of the students and awareness about Saudi Stem Cells Donor Registry (n = 217).**

| Variable | | Awareness about Saudi Stem Cells Donor Registry | | |
|---|---|---|---|---|
| | Total | Yes | No | p value |
| Less than 21 years | 62 | 30 (48.4) | 32 (51.6) | 0.796 |
| 21–22 years | 86 | 46 (53.50) | 40 (46.5) | |
| Above 22 years | 69 | 34 (49.3) | 35 (50.7) | |
| Gender | | | | |
| Male | 118 | 48 (40.7) | 70 (59.3) | 0.020[*] |
| Female | 99 | 62 (62.6) | 37 (37.4) | |
| College type | | | | |
| Medical | 133 | 70 (52.6) | 63 (47.4) | 0.489 |
| Dental | 84 | 40 (47.6) | 44 (52.4) | |
| Year of education | | | | |
| 1st | 55 | 28 (50.9) | 27 (49.1) | 0.102 |
| 2nd | 48 | 20 (41.7) | 28 (58.3) | |
| 3rd | 59 | 29 (49.2) | 30 (50.8) | |
| 4th | 28 | 13 (46.4) | 15 (53.6) | |
| 5th | 27 | 20 (74.1) | 7 (25.9) | |
| Marital status | | | | |
| Single | 201 | 105 (52.20) | 96 (47.8) | 0.124 |
| Married | 16 | 5 (31.3) | 11 (68.8) | |
| Smoking status | | | | |
| Daily | 27 | 7 (25.9) | 20 (74.1) | 0.002[*] |
| Rarely | 28 | 10 (35.7) | 18 (64.3) | |
| Never | 162 | 93 (57.4) | 69 (42.6) | |

**Notes.**
*Statistically significant at the level of 0.05 (Chi-square test).

which they found that 76.5% had fair knowledge. Majority of the other studies done in Saudi Arabia have noted poor knowledge of the participants regarding the stem cells. Alhadlaq et al. in their study regarding implications of stem cells in dentistry observed poor knowledge among the male and females students (*Alhadlaq et al., 2019*). *Jawdat et al. (2018)* in their study in general population observed poor knowledge about Cord Blood banking with 66% of participants possessing inadequate knowledge. *Azzazy & Mohamed (2016)* in their study on nursing students found poor knowledge about stem cells therapy which however remarkably improved after educational intervention. *AlAbdulqader et al. (2017)* noted overall poor knowledge among the general population about the stem cells. Poor knowledge in these studies may be due to the fact that most of these studies were done on general population. Better knowledge levels in our study may be attributed to the fact that stem cells are being widely used in the Kingdom of Saudi Arabia to treat a number of diseases as well as the fact that we dealt with the Medical and dental students who are future health care professionals.

Regarding attitude towards stem cells, we noted majority (70%) of the students possessed high attitude score and 18.9% had excellent attitude score (Table 2). The high level

of positive attitude towards stem cells may be attributed to medium to high level of knowledge noted among the students in our study. Lye et al. observed in their study that 76.1% of the nursing students possessed good attitude towards stem cell application in medical science (*Lye et al., 2015*). Many studies done in Saudi Arabia have found positive attitude of the study participants towards the stem cells and their implications in medicine. *ALtemani et al. (2018)* noted fair attitude regarding stem cell transplantation in 73.1% of their participants. A moderately positive attitude was also observed by *Alhadlaq et al. (2019)* in their study. *AlAbdulqader et al. (2017)* however noted poor attitude of the population about the stem cells. *Venugopal et al. (2016)* observed neutral attitude (78.6%) among the nurses regarding stem cells and umbilical cord blood banking with a mean attitude score of $53.75 \pm 8.26$ (*Venugopal et al., 2016*).

In our study, 50.7% of the students were aware about the SSCDR stem cell donor registry (Table 1). *Zaini & Al-Thagafi (2020)* in their study on medical students in Taif university found that 52.6% of the students had never heard about the SSCDR whereas only 5.6% of participants were registered in SSCDR. A significant relationship was noted between the SSCDR awareness and knowledge score but when compared with attitude score the relationship was insignificant (Table 3). We did not find any significant relationship of the socio-demographic characteristics with either the knowledge or the attitude scores (Table 3). *ALtemani et al. (2018)* did not find significant statistical difference across gender, between doctors and medical students regarding the knowledge, attitude, practice, and the total score. Significant difference in the knowledge score between male and female participants was observed by Alhadlaq et al. However regarding the attitude score, significant differences were not noted between males and females (*Alhadlaq et al., 2019*). *AlAbdulqader et al. (2017)* in their study regarding knowledge about stem cells observed significant role of age, level of educational and specialty of the participants. However they failed to find significant contribution of the above mentioned criteria as far as attitude of the participants towards stem cells was concerned.

A significant positive correlation with $r = 0.334$, $p < 0.001$ was observed between the knowledge and attitude scores in our study. Similarly, a significant positive linear correlation was noted by Alhadlaq et al. between knowledge and attitude for males ($r = 0.323$, $p = 0.00$) and females ($r = 0.392$, $p = 0.00$). *Venugopal et al. (2016)* also noted a significant positive correlation ($r = 0.532$; $p < 0.01$) between the knowledge score and attitude score in their study on nurses regarding stem cells and umbilical cord blood banking (*Venugopal et al., 2016*). However, *Lye et al. (2015)* noted poor correlation between the knowledge and attitude score among the study participants ($r = 0.08$). They also observed a $p$ value $>0.05$ thereby indicating insignificant correlation between the years of education and the knowledge score. *AlAbdulqader et al. (2017)* in their study observed weak correlation between knowledge and attitude.

A significant association was observed between the SSCDR awareness and gender of the participants ($p$-value of 0.020) and their smoking status ($p$-value of 0.002) (Table 4). To the best of our knowledge, there was no published study in local settings available that attempted to find the association between socio-demographic characteristics and awareness about SSCDR. We recommend health educational programs to increase the awareness regarding

the SSCDR and its activities, which in turn will increase the knowledge and improve the attitude of the medical students towards stem cells usage.

## CONCLUSION

Majority of the students possessed medium to high knowledge, whereas 70% of the students possessed high attitude score towards stem cell research and its medical significance. A significant relationship was observed between Stem cell Donor Registry Awareness and the knowledge score. Significantly positive correlation between the knowledge and attitude scores was also noted. However relationship of the various demographic characteristics with the knowledge and attitude scores was not statistically significant.

It is proposed to include various interventional educational programs on the importance of stem cells in the medical field taking into account the religious, ethical, moral, cultural, and social factors. This becomes imperative in view of the importance of stem cells in the present medical scenario as well as the fact that the medical and dental students are the future health care providers. This will help the students to gain high to excellent levels of knowledge thereby increasing the positive attitude levels in the students which will be extremely beneficial in imparting proper information regarding the stem cells to the patients and general population.

### Strength of the study

This research will identify the knowledge and attitude of health science students towards stem cells and their applications in medical science. Increasing their knowledge and positive attitude will influence the local community as many times, they may be the first level contact of any health needs of people. This research may give an opportunity to implement the health education program of the Saudi Center for Organ Transplantation in health science colleges.

### Limitations of the study

Firstly, the study is a cross-sectional study which identifies the association between variables, and not the causation and its direction. Secondly, the study has self-reported data only and has limitations such as subjectivity, exaggerated reports and recall bias. Finally, this research was conducted across colleges within Jouf University. This research could have been enhanced further by including health colleges from other universities in the KSA.

### Funding

The authors received no funding for this work.

### Competing Interests

The authors declare there are no competing interests.

## Author Contributions

- Abdulrahman Almaeen conceived and designed the experiments, authored or reviewed drafts of the paper, and approved the final draft.
- Farooq Ahmed Wani conceived and designed the experiments, performed the experiments, authored or reviewed drafts of the paper, and approved the final draft.
- Ashokkumar Thirunavukkarasu conceived and designed the experiments, performed the experiments, analyzed the data, prepared figures and/or tables, and approved the final draft.

## Human Ethics

The following information was supplied relating to ethical approvals (i.e., approving body and any reference numbers):

The Local Committee for Bioethics (LCBE) of Jouf University approved this research (approval no: 09-02/41).

## Data Availability

Raw data are available in the Supplemental File.

## Supplemental Information

Supplemental information for this article can be found online at http://dx.doi.org/10.7717/peerj.10661#supplemental-information.

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
