# Peer review of "Knowledge and attitudes towards stem cells and the significance of their medical application among healthcare sciences students of Jouf University"

_PeerJ, doi:10.7717/peerj.10661_

## Round 0.1 · original submission · Major Revisions

Your manuscript has been reviewed and requires modifications prior to making a decision. The comments of the reviewers are included at the bottom of this letter. Reviewers indicated that the introduction, methods, and results sections should be substantially improved. I agree with the evaluation and I would, therefore, request for the manuscript to be revised accordingly. I would also like to suggest the following changes:

• Please use a normality test to check the normality assumption of the data? Because to use Pearson correlation, ANOVA, and t-test methods, the normality assumption of the data should be met.
• The authors did not mention ANOVA and t-test methods in the statistical analysis section.
• The reliability analysis of the scale is not evaluated. Please provide Cronbach alpha values of the original and applied scale.
• The sample size calculation formula should be written in scientific representation.
• Please get biostatistical consulting to improve the manuscript.

Reviewer 1 ·

Basic reporting

no comment

Experimental design

no comment

Validity of the findings

no comment

Additional comments

In this manuscript, the authors aim to assess the knowledge and attitude of the medical and dental students towards the stem cells and their applications in medical science. The manuscript is clearly written and it is dealing with an interesting topic.
The authors may consider the following comments:
Line 120: please, make sure that the reference Browne 1995 ("On the use of a pilot sample for sample size determination") is adequate to explain the knowledge and awareness among public and healthcare staff about CB banking.
Section Statistical Analysis: in the description of the statistical methods, add the ANOVA test and the independent t-test that you used for comparison of the mean values of knowledge and attitude scores according to socio-demographic characteristics of the sample, as indicate in Table 3.
Lines 187 and 217: Percentage of married students is reported as 7.2, while in Table 1 as 7.4. Please, resolve this inconsistency.
Line 208: it's unclear the meaning of the words "do. Published 2020". Please, clarify the meaning. If it refers to a work unpublished, cite it as "unpublished data" and supply the author's first initial and surname, and the year of the data collection, as indicated in the "Instructions for the authors".
Lines 238, 252, 266, 274: please complete the reference "AlAbdulqader et al." with publication year.
Line 301: complete the reference with publication year, full title of the Journal, volume: page extents.
Line 319: it's unclear the reference "do. ScWtaawt. Published 2020". Please, clarify.

Reviewer 2 ·

Basic reporting

No comment

Experimental design

No comment

Validity of the findings

No comment

Additional comments

This is an interesting study and the authors have collected a unique dataset using cutting
edge methodology. The paper is generally well written and structured. However, in my opinion the
paper has some shortcomings in regards to some data analyses and text, and I feel this unique dataset
has not been utilized to its full extent. Below I have provided numerous remarks on the text as it is
often vague and long-winded

In the abstract:
You need to avoid "We" and try to use passive voice.
You need to delete some of the methods part and rewrite it for example no need to mention the statistical analysis part in the abstract.
The abstract was copied from the text of the article, which needs some modification.

Methodology section:
In the methodology and sample size calculation, the formula was used contain "d2", the margin of error e2 and also the authors need to write the name of this formula "cochrane". and referred p to proportion will be better than prevalence.
The authors, need to mention the sample size method they use it (i.e. stratified or random,..etc)
In the ethical part "Jouf university" the first letter should be capital.
In the statistical analysis section:
Noting mentioned about hypothetical test such as ANOVA and independent t test but both of them presented in table3.

In the result section:
No need for figure 1, it can be added to the table 1.
No need for table 4, the sentence already mentioned in the text is enough.

The authors need to do some more analysis to assess the awareness of the participant about Stem Cell Donor Registry and their year of study or gender (cross tab and Chi-square), also to add chi square to analysis section of the methodology.

Discussion:
Nothing mentioned about the limitation of the study, the authors need to mention some limitations to be avoided in the future studies, and in some part I feel like it is a rewritten of the result section.

In line number 229, the authors mentioned "Majority of the other studies.." and only one reference was used, what about the other studies!
The authors compare their results with other studies were done in KSA, why no international study was mentioned?

Conclusion:
Need to rewrite to avoid active statement "We" and use passive voice. Also, some parts of the conclusion was written like a result not conclusion i.e. p- value, these sentences should be deleted.

---

## Round 0.2 · Minor Revisions

The manuscript has been assessed by two reviewers, and one of them agrees with the fact that there are still a few points that need to be addressed. We would be glad to consider a substantial revision of your work, where the second reviewer’s comments will be carefully addressed one by one.

I would also like to suggest the following changes:

• Please give the Kolmogorov-Smirnov or the Shapiro-Wilk normality test results.
• The total number of students should be included in the abstract.
• Please give the Cronbach alpha values for all data (not for only pilot study).

Reviewer 1 ·

Basic reporting

no comment

Experimental design

no comment

Validity of the findings

no comment

Additional comments

The authors responded satisfactorily to my comments.

Reviewer 2 ·

Basic reporting

None

Experimental design

None

Validity of the findings

None

Additional comments

I want to thank the authors for their response to my comments, most of my comments taken in consideration, I want to add:

In cronbach alpha as the authors calculated from pilot study to be more than 0.7, it is better to add a comment like "which exhibited good internal consistency in the format used" or any other comment.

The authors calculated the sample size two times, one is enough.

The authors used skewness and kurtosis to test the normality, using, Kolmogorov-Smirnov or Shapiro are better depends on the sample size in each group.

The authors wrote they used ANOVA to test the association..., for association Chi square test is used, ANOVA used to compare between group as a hypothetical test.

Instead of "proforma" word, instrument is better and more usable, same with "question" statement or item is better.

In table 3, attitude section Mean+-SD not mean (SD),year of education 4th should add +-, smoking status for attitude need -, under table 4 please write the test used and statistically significant at which level 0.01 or 0.05 for example. Please review the tables for any error.

Other than that, I don't have any more comment and I want to than the authors again.

---

## Round 0.3 · accepted · Accept

The authors addressed the reviewer's concerns and substantially improved the content of MS. So, based on my own assessment as an academic editor, no further revisions are required and the MS can be accepted in its current form.

Reviewer 2 ·

Basic reporting

-

Experimental design

-

Validity of the findings

-

Additional comments

I want to thank the authors for their responses to my comments, the manuscript is very well written; clear, precise, and easy to understand. For me, all comments were taken in consideration and now it is ACCEPTABLE.